# Soil Mesofauna Responses to Fire Severity in a Sclerophyllous Forest in Central Chile

Ricardo Castro-Huerta [1] , Carolina Morales [1], John Gajardo [2,*], Enrique A. Mundaca [1] and Marco Yáñez [3]

1. Escuela de Agronomía, Facultad de Cs. Agrarias y Forestales, Universidad Católica del Maule, Maule 3466706, Chile; rcastroh@ucm.cl (R.C.-H.); carolina.mora.saave@gmail.com (C.M.); emundaca@ucm.cl (E.A.M.)
2. Instituto de Bosques y Sociedad, Facultad de Ciencias Forestales y Recursos Naturales, Campus Isla Teja, Universidad Austral de Chile, Valdivia 5090000, Chile
3. Instituto de Investigación Interdisciplinaria, Universidad de Talca, 2 Norte 685, Talca 3460000, Chile; marcoyanez@utalca.cl
* Correspondence: john.gajardo@uach.cl

**Abstract:** Forest fires may have severe impacts on the aboveground biodiversity and soil chemical and biological properties. Edaphic organisms are highly sensitive to disturbances and are typically used to measure the magnitude of these events. Overall, little is known about the responses of these organisms to fires differing in their severity levels. This study aimed to assess the effect of fire severity on the soil mesofauna community diversity and structure in a site located in a Mediterranean zone of central Chile. In postfire conditions, we use spectral indexes from satellite images to map fire severity at four levels (non-damage (ND), low damage (L), medium damage (M), high damage (H)). Soil samples were collected at each severity level, and the mesofauna abundance was quantified. Although the metrics describing species diversity and dominance were similar among fire severity levels, the abundance and composition of the mesofauna were specifically altered at the high severity level. The edaphic mesofauna can be considered suitable bioindicators to evaluate the postfire ecosystem recovery, especially in the areas highly damaged by fire.

**Keywords:** arthropods; diversity; ecosystem recovery; Sentinel 2

## 1. Introduction

Soils contain a wide diversity of organisms affecting the earth's biogeochemical processes and consequently the ecosystem functioning [1–3]. Among these organisms, the edaphic fauna is classified according to adult width size in microfauna (<0.2 mm), mesofauna (0.2–2 mm), and macrofauna (<2 mm) [4,5]. Overall, most of the meso- and macrofauna correspond to arthropods, which are represented by Isopoda, Myriapoda, Insecta, Acari and Collembola, as inhabitants of the litter and the soil [6].

Anthropogenic and natural disturbances can change the composition and abundance of the arthropod communities [7,8], which is why the edaphic fauna is considered as a biological indicator of soil health [9–11]. Forest fires are one of the most important large-scale disturbances affecting an ecosystem's functioning [12]. Fires can reduce the vegetation strata and biodiversity and increase the soil degradation and emission of greenhouse gases, whose magnitude is associated with fire severity [13]. Fire severity maps provide a spatial quantification of a fire's impact on the landscape and can be used to prioritize resources and as baselines for future monitoring requirements [14]. In this context, the fire severity measured the ecological changes that have occurred in a burned area (postfire) relative to the previous conditions (prefire), which depends on the duration and intensity of fires [15]. In the end, the soil biological and physicochemical properties, postfire climatic conditions, and the depth of the horizontal fire radiation determine the resilience of the soil communities [16] associated with both the survival patterns and colonization of soil organisms [17–20].

In January and February 2017, the Mediterranean zone of Chile was affected by one of the most extensive fires in its modern history, covering an area of approximately 587,000 hectares, mainly of sclerophyllous forest and commercial plantations, adding a new level to the world scale used to classify forest fires [21]. Although these large fires are uncommon in this zone [22], current trends show an increase in the frequency of these fire events in the world [23]. Satellite technologies allow a fairly accurate modelling of fire severity by quantifying differences in plant biomass [24–26], and the accuracy is still increasing with new developments [27–30]. These technologies have even been successfully applied to estimate fire severity in arid areas [31] and agricultural crops [32], but their application needs to be validated and compared with real data [33–35]. Overall, there are concerns about the impacts of fires on the global ecosystem, especially on soil invertebrates, which provide essential ecosystem services and are crucial to energy flow and nutrient cycling processes [36,37]. Fontúrbel et al. [38] mentioned that soil mesofauna may need more time to recover post fire compared with other soil organisms and that a fire may change the structure of the soil organic layer, which is the habitat of these communities. Thus, in this study, we assessed the responses of the soil arthropod mesofauna to the fire severity in a typical sclerophyllous forest found in the Mediterranean zone of Chile. We use satellite images and Geographic Information Systems technology to identify areas with different levels of fire severity and sampled the soil arthropod mesofauna composition and diversity present in these areas. We hypothesize that the postfire soil mesofauna community structure is negatively altered when increasing the fire severity.

## 2. Materials and Methods

### 2.1. Study Site and Soil Characteristics

This study was carried out in the Central Valley of Chile, specifically in the area of Villavicencio, San Javier district, Maule Region [Figure 1]. In January 2017, this area was affected by one of the biggest fires in modern Chilean history [39]. The site is located at an average altitude of 158 m.a.s.l. and has a temperate mesothermal climate, Mediterranean stenothermal semiarid inland, and annual precipitation of 840 mm. Mean temperatures vary from 30.1 °C in summer to 4 °C in winter [40]. The vegetation corresponds to a sclerophyllous and deciduous forest [41]. The state is a regressive climax represented by a degraded open forest and shrubland, with some dominance of invasive species, which undergo rapid growth and regeneration under recurrent fires [42]. The soil belongs to Pocillas soil series, characterized by sedimentary soil of lacustrine origin, loamy clay texture, grey to brown color, and 40–50 cm depth, pH 6.5, and 3.7% organic matter [43].

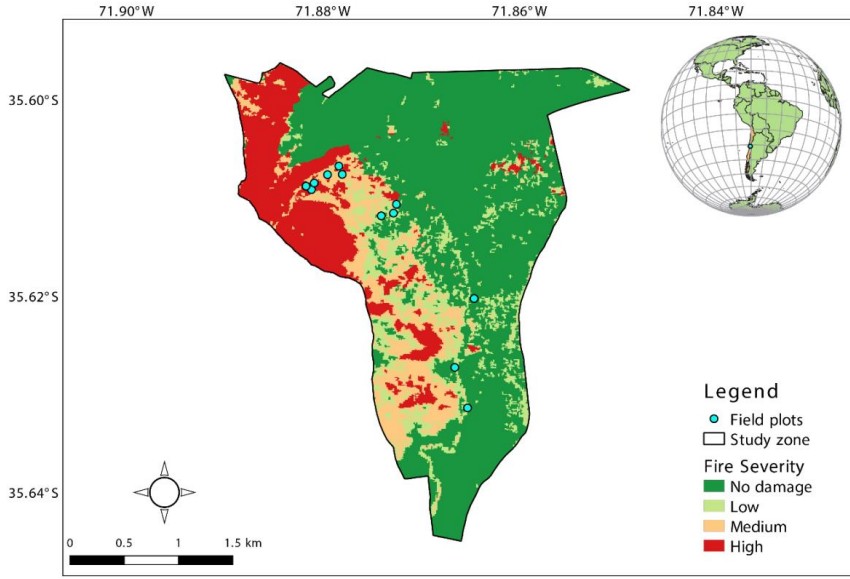

**Figure 1.** Villavicencio study site, commune of Villa Alegre Region del Maule, Chile.

### 2.2. Remote Sensing Data and Experimental Design

Fire severity was estimated using data from the Multispectral Instrument [MSI] sensor onboard the Sentinel-2 satellite platform [44], whose data is freely available and can be downloaded from the Copernicus Open Access Hub [45]. Based on the short-term severity assessment indicated by Key & Benson [24], two summer images were used, before (19 January 2017) and after (20 March 2017) the fire. The satellite data were atmospherically corrected to obtain surface reflectance using the Sen2Cor algorithm [46]. Considering the field plot size, we took advantage of the high spatial resolution of the MSI sensor and decided to resample the bands used at 10 m. The spectral indices Normalized Burn Ratio [NBR], difference Normalized Burn Ratio [dNBR] [15], and Relative delta Normalized Burn Ratio [RdNBR] [25] were calculated using the satellite images.

$$\text{NBR} = \left( \frac{\rho\text{NIR} - \rho\text{SWIR}}{\rho\text{NIR} + \rho\text{SWIR}} \right) \times 1000 \tag{1}$$

$$\text{dNBR} = \text{NBR}_{\text{PRE}} - \text{NBR}_{\text{POST}} \tag{2}$$

$$\text{RdNBR} = \frac{\text{dNBR}}{\sqrt{\left| \frac{\text{NBR}_{\text{PRE}}}{1000} \right|}} \tag{3}$$

where, $\rho\text{NIR}$, $\rho\text{SWIR}$, are the surface reflectance for the near infra-red band [B8] and the short-wave near infra-red band [B12]; $\text{NBR}_{\text{PRE}}$ and $\text{NBR}_{\text{POST}}$ represent the NBR for the situations previous to and after the fire, respectively. According to the thresholds obtained from multiple fires and regions by Parks et al. [30], calculated RdNBR continuous values were discretized in four severity categories: No damage (ND), low (L), medium (M), and high (H). Satellite images were processed by QGIS 2.18 software [47].

### 2.3. Soil Sampling and Processing

On November 30, 2017, three sampling plots per severity level were randomly distributed [Figure 1]. We took care of locating the plots on homogeneous patches within the severity levels, privileging accessible sectors, and avoiding steeped slopes to guarantee personnel safety [33]. To allow locating the plot under homogenenious condition, we computed the Soil Adjusted Vegetation (SAVI) [48] from pre-fire images. SAVI values varied from 0.17 to 0.35, which indicates the open-areas within the site. We considered these values relatively homogeneous regarding the degradation stage of this ecosystems. The sample collection was carried out according to the modified Tropical Soil Biology and Fertility protocol [49], based on the extraction of a 10 × 10 × 10 cm soil pit, designed to collect endogenic fauna. Subsequently, the samples were placed in labeled bags and subjected to a mesofauna extraction system two days after collect using the Berlese-Tüllgren method to ensure high extraction efficiency. [50]. Briefly, it consisted of a funnel with a sieve [opening diameter of 2 mm or 10 mesh], where the soil was unbundled manually and located inside the funnel under 60 W light bulb for seven days to ensure the mesofauna fall into a flask containing a 70% alcohol solution.

### 2.4. Species Identification and Biodiversity Analysis

Species identification was assessed at each sample through a binocular stereoscope loupe (Olympus® SZ40 with light source TL2, Lytle, TX, USA), and using the Recognizable Taxonomic Unit (RTU) criterion for each taxonomic group indicated in Table 1. This was used to separate unknown taxa and assess species richness when identification at species level was not possible [51]. The Shannon-Wiener and Simpson diversity indexes were calculated per sample. The former assumes that all species are represented in the samples, indicating uniformity in abundance according to all species sampled [52]. The latter focuses on the most abundant or dominant species, estimating the probability that two individuals randomly sampled from the same sample belong to the same species [53].

**Table 1.** Total abundance of soil arthropods mesofauna per species.

| Soil Arthropods Mesofauna | Severity Treatment | | | | | | | | | | | |
|---|---|---|---|---|---|---|---|---|---|---|---|---|
| | No Damage | | | Low | | | Medium | | | High | | |
| | RTUs | Abundance | % | RTUs | Abundance | % | RTUs | Abundance | % | RTUs | Abundance | % |
| Arachnida | | | | | | | | | | | | |
| Acari | | | | | | | | | | | | |
| Sarcoptiformes | 6 | 62 | 9.27 | 7 | 20 | 2.79 | 3 | 14 | 3.36 | 1 | 1 | 0.70 |
| Mesostigmata | 9 | 137 | 20.48 | 9 | 90 | 12.55 | 7 | 67 | 16.07 | 6 | 33 | 23.24 |
| Oribatida | 18 | 288 | 43.05 | 21 | 252 | 35.15 | 24 | 207 | 49.64 | 7 | 23 | 16.20 |
| Trombidiformes | 5 | 6 | 0.90 | 5 | 36 | 5.02 | 6 | 29 | 6.95 | 4 | 33 | 23.24 |
| Other Acari | 2 | 48 | 7.17 | 2 | 21 | 2.93 | 2 | 7 | 1.68 | 7 | 9 | 6.34 |
| Araneae | 1 | 2 | 0.30 | 1 | 1 | 0.14 | 0 | 0 | 0 | 0 | 0 | 0 |
| Pseudoscorpionida | 1 | 8 | 1.20 | 1 | 7 | 0.98 | 1 | 15 | 3.60 | 0 | 0 | 0 |
| Hexapoda | | | | | | | | | | | | |
| Ectognatha | | | | | | | | | | | | |
| Coleoptera | 3 | 20 | 2.99 | 1 | 4 | 0.56 | 3 | 4 | 0.96 | 3 | 7 | 4.93 |
| Diptera | 3 | 13 | 1.94 | 4 | 76 | 10.60 | 2 | 4 | 0.96 | 0 | 0 | 0 |
| Hemiptera | 0 | 0 | 0 | 1 | 1 | 0.14 | 2 | 2 | 0.48 | 0 | 0 | 0 |
| Hymenoptera | 0 | 0 | 0 | 0 | 0 | 0 | 0 | 0 | 0.00 | 1 | 2 | 1.41 |
| Isopoda | 0 | 0 | 0 | 1 | 1 | 0.14 | 1 | 1 | 0.24 | 0 | 0 | 0 |
| Lepidoptera | 1 | 1 | 0.15 | 1 | 1 | 0.14 | 0 | 0 | 0.00 | 1 | 1 | 0.70 |
| Protura | 0 | 0 | 0 | 1 | 37 | 5.16 | 1 | 7 | 1.68 | 1 | 1 | 0.70 |
| Psocoptera | 3 | 4 | 0.60 | 1 | 1 | 0.14 | 2 | 8 | 1.92 | 3 | 10 | 7.04 |
| Thysanoptera | 1 | 1 | 0.15 | 0 | 0 | 0 | 1 | 1 | 0.24 | 1 | 3 | 2.11 |
| Entognatha | | | | | | | | | | | | |
| Collembola | 9 | 45 | 6.73 | 19 | 137 | 19.11 | 6 | 37 | 8.87 | 5 | 19 | 13.38 |
| Diplura | 2 | 9 | 1.35 | 1 | 4 | 0.56 | 1 | 2 | 0.48 | 0 | 0 | 0 |
| Myriapoda | | | | | | | | | | | | |
| Chilopoda | 1 | 17 | 2.54 | 1 | 11 | 1.53 | 1 | 12 | 2.88 | 0 | 0 | 0 |
| Symphyla | 1 | 8 | 1.20 | 2 | 17 | 2.37 | 0 | 0 | 0 | 0 | 0 | 0 |
| Total | 66 | 669 | 100 | 79 | 717 | 100 | 63 | 417 | 100 | 40 | 142 | 100 |

*2.5. Statistical Analysis*

We analysed the degree of structuring per sample unit using a non-metric Multidimensional Scaling [nmMDS] analysis [54,55] to calculate the Bray–Curtis and Jaccard similarity matrix [53]. Clusters were determined by adjusting convex envelope graphics [Convex Hull] and the calculation of the Kruskal Stress value, which is considered adequate to define groupings at values lower than 0.1, and unacceptable when above 0.15 [56,57]. To identify differences among clusters a one-way similarity analysis [ANOSIM] was used based on the distance measures of the Bray–Curtis [54] and Jaccard index [53]. These analyses were performed using the free software PAST® (Oslo, Norway) [58]. Measures of diversity and composition were analysed through a one-way analysis of variance using the linear models and generalized linear models for continuous and discrete variables, whereas mean comparisons was determined by the Tukey's test, using R (www.r-project.org (accessed on 20 June 2021)). Significant differences were considered at a probability level of 0.1 regarding the low replication of the study.

## 3. Results and Discussion

This study partially supports the hypothesis that the postfire soil mesofauna community structure is negatively altered when increasing the fire severity. The multidimensional scaling analysis showed a clear distinction between the groups established with the Jaccard's method (Kruskal's stress value of 0.11) (Figure 2), mainly due to a differentiation of the high severity level (H) relative to the others. Similarly, the one-way similarity analysis (ANOSIM) indicated significant differences among fire severity levels on the Jaccard index (R = 0.3812, $p$ = 0.0167), implying differences in the community composition [59], and consequently an effect of fire on the habitat of soil invertebrate communities [20]. Otherwise, the Bray–Curtis analysis did not showed differences between severity levels (Kruskal's stress value of 0.06, R = 0.084, $p$ = 0.2970). Overall, the results must be interpreted with care because of the low replication of this study. Even so, the pairwise comparison among severity levels revealed some evidence about the differences in community composition in the high severity level relative to the other levels ($p$ = 0.1) [Table 2].

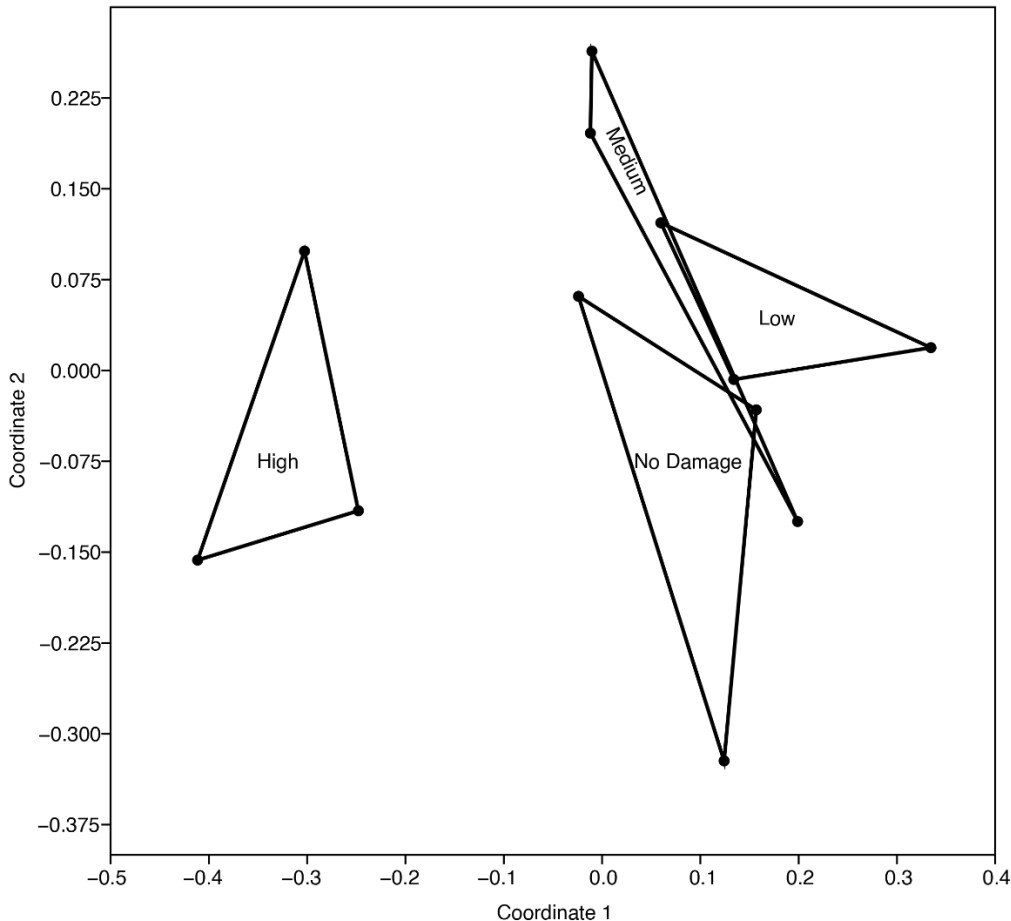

**Figure 2.** Jaccard similarity grouping, Kruskal stress = 0.11.

**Table 2.** R test and *p*-value from the analysis of similarities (ANOSIM) for paired comparison among fire severity levels.

| | Bray–Curtis | | Jaccard | |
|---|---|---|---|---|
| **Paired Comparison** | **R** | ***p*-Value** | **R** | ***p*-Value** |
| ND vs. L | −0.2593 | 0.7986 | −0.0741 | 0.7973 |
| ND vs. M | −0.0741 | 0.6978 | 0.0020 | 0.5992 |
| ND vs. H | 0.5185 | 0.0989 | 0.7407 | 0.1031 |
| L vs. M | −0.2222 | 0.8034 | 0.0001 | 0.4057 |
| L vs. H | 0.4444 | 0.7986 | 0.9630 | 0.0995 |
| M vs. H | 0.2963 | 0.2018 | 0.8704 | 0.0988 |

Table 3 shows the mean values of species richness, relative abundance, Shannon-Wiener, and Simpson biodiversity, which are based on the proportional abundance of the quantification of species to estimate diversity [52,60]. Overall, all the parameters were relatively similar among the fire severity level, except abundance, which exhibited lower values in the high severity level. The low species richness in high severity is probably a consequence of decreased vertebrate survival and a slow recolonization process. The mean Shannon-Weaver index and Evenness among severity levels varied from 1.65 to 1.81 and 0.51 to 0.66, respectively. According to Pla [61], these values indicate a high diversity and homogeneity in the distribution of the mesofaunal community. Unlike, some studies show that a higher fire severity may increase the dominance, since there are populations of invertebrates that are favored by fires [62–64].

**Table 3.** Mean (±standard error) for diversity and composition structure parameters by fire severity level.

| Severity Level | Taxa | Abundance | Dominance | Simpson Index | Shannon Index | Evenness |
|---|---|---|---|---|---|---|
| ND | 11.0 ± 1.52 a | 222.3 ± 108.0 a | 0.23 ± 0.06 a | 0.77 ± 0.06 a | 1.81 ± 0.18 a | 0.60 ± 0.13 a |
| L | 10.6 ± 1.76 a | 239.0 ± 106.0 a | 0.26 ± 0.04 a | 0.73 ± 0.04 a | 1.69 ± 0.10 a | 0.54 ± 0.09 a |
| M | 10.6 ± 1.20 a | 139.0 ± 61.8 a | 0.29 ± 0.07 a | 0.71 ± 0.07 a | 1.66 ± 0.18 a | 0.52 ± 0.08 a |
| H | 8.33 ± 1.33 a | 47.3 ± 14.3 b | 0.25 ± 0.08 a | 0.74 ± 0.08 a | 1.65 ± 0.24 a | 0.66 ± 0.11 a |

Note: Different letters indicate significant differences according to the Tukey's comparisons test.

Remote Sensing tools used to estimate severity are based on the response of the reflectance of objects affected by the fire on the Earth's surface. When the estimated severity level is high, it generally implies the removal of the entire vegetation cover, the response being mainly from the soil [13]. However, when the severity level is medium or low, optical sensors can only observe responses from the upper parts of the canopy. Other phenomena that occur in the lower strata, including the ground, may be invisible from the perspective of satellites [25]. This combination of factors affects the adequate quantification of severity [15]. The latter could explain that the main differences between the communities found in the different levels of severity occurred only in categories H and ND.

A total of 139 arthropod RTUs were recorded from all samples (Table 1). The most frequent and abundant taxa among all samples were Collembolla and Acari. This result is not surprising as both groups have been reported to constitute approximately 72% to 98% of the soil arthropod fauna, the most representative and abundant groups of the edaphic mesofauna [65]. In this regard, these groups of organisms are known to regulate and stabilize the soil through a complex network of interactions, participating in the decomposition of organic matter and the cycling of nutrients [66–68]. In terms of functional groups, we found particular arthropod groups to occur in specific fire severity levels (Table 1). For instance, a high amount of Oribatida mites were found in all the severity levels, except in H. This finding is likely explained by a higher loss of soil organic matter in high severity because these organisms are primarily detritivorous and secondary decomposers and play a crucial role in transforming organic matter [66,67,69]. Otherwise, the higher presence of Oribatida is not surprising as it is known to dominate nutrient-poor soils [70]. Thus, a medium severity level might alter the nutrient status in these soils. Oppositely, Trombidiformes tended to be higher in their severity level.

Mesostigmata mites were among the most abundant organism. These are known for being predators of the micro- and mesofauna and as soil quality indicators since they commonly show high population numbers in undisturbed soils [71]. In this study, there was no abundance pattern of Mesostigmata mites associated with the fire severity, which agreed with the study by Kamczyc et al. [72]. At high severity, there was a higher abundance of Psocoptera, a pioneering group colonizing disturbed areas, whose presence indicates a soil recovery action [11]. In addition, in areas with a high impact of fires, soil nutrients are abruptly mineralized, decreasing food sources for soil invertebrates and slowing their recovery [73]. On the other hand, the high abundance of Collembolla at low severity (L) is a good indicator of healthy in soils, as collembolans feed on saprofitic fungal hyphal networks which are particularly susceptible to disturbance and contribute to the control of phytopathogenic microorganisms in agriculture systems. [5]. In general, in sclerophyllous ecosystems such as the one in this study, it is common to observe a high variability in plants and fauna, which allows a rapid recovery of the ecosystem providing that the soil is not devoid of vegetation [62]. Thus, the maintenance of diversity and ecological processes depends on the heterogeneity and structural complexity [74].

## 4. Conclusions

In the assessed sclerophyllous forest, the postfire soil mesofauna community structure was altered only with a high fire severity level. Thus, we partially accept our hypothesis as the results showed no effect on the soil mesofauna communities at low and medium fire severity, suggesting a high resilience of the edaphic richness at these levels of damage.

Furthermore, the ability to remotely assess the damages of forest fires would considerably reduce the costs and time required to evaluate the actual damage in the field. However, the estimation of postfire damages with remote sensing tools must be validated with biological data to improve their estimation when projecting reforestation and restoration strategies.

**Author Contributions:** Conceptualization, R.C.-H., C.M. and E.A.M.; methodology, R.C.-H., C.M., E.A.M. and J.G.; writing—original draft preparation, R.C.-H., C.M., E.A.M., J.G. and M.Y.; writing—review and editing, R.C.-H., C.M., E.A.M., J.G. and M.Y. All authors have read and agreed to the published version of the manuscript.

**Funding:** This research was funded by the Fondo de Investigación del Bosque Nativo (FIBN) de la Corporación Nacional Forestal de Chile (CONAF), grant number FIBN 010/2017.

**Acknowledgments:** We would like to thanks to Baron Philippe de Rothschild, a vineyard company, who provided the experimental site for this study.

**Conflicts of Interest:** The authors declare no conflict of interest. The funders had no role in the design of the study; in the collection, analyses, or interpretation of data; in the writing of the manuscript, or in the decision to publish the results.

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
