# Peer review of "Soil Mesofauna Responses to Fire Severity in a Sclerophyllous Forest in Central Chile"

_forests, doi:10.3390/f12111444_

Round 1

Reviewer 1 Report

Castro-Huerta et al. present an interesting observational study of how forest fires impact soil and litter dwelling arthropods. They were able to take advantage of a forest fire in 2017 to investigate the impacts of the fire, and explored how communities of several taxa responded to the disturbance. I think the manuscript has a good foundation, but some work can be done to present a stronger study using the data the authors have in hand. I have included my comments section-by-section below, denoted using the line numbering.

Introduction:

Post-line 46: I think a paragraph here (before the design lead-in) covering how soil organisms and fire disturbances are linked would help to better lay out the story the authors are presenting with this paper. Right now it is unclear why the organisms selected for study are being used and why one might expect them to respond to the fire disturbance being studied.

Methods:

Lines 106-108: The authors note that sites within severity levels were randomly located on homogeneous patches. How similar were sites between the severity levels? Do data exist that can show how the areas affected by the burn related to one another prior to the disturbance? In order to be able to demonstrate that any observed differences are due to the fire severity, it is imperative that the authors be able to demonstrate that prior conditions at the site were similar enough that they are unlikely to be driving the observed patterns. Do vegetation and soil physical/chemical data exist (or that could be inferred) to be compared?

Section 2.5: I would like to see a power analysis run on the design in order to have a better idea what the statistical power of the experiment is.

Lines 134-136: I would like to see comparisons of the diversity indexes that were calculated. There is a good amount of data presented in Table 2, but none of it was subjected to statistical testing for use in the study, which is a shame.

Results and Discussion:

I think the section would benefit from a change in the order data are presented and discussed. I suggest organizing the section by fire severity/disturbance level, which would minimize the repetition currently in the text due to grouping via taxon. Right now the split by taxon leads to repeated mention of high or low quality soil resources, which makes the text a bit more difficult to follow. Framing the section by fire severity/disturbance level would also lead to this section better mirroring the format of the introduction and keeping the story more consistent in its presentation.

Table 1: Italicizing results found to be significant in this study would help readers to pick up the presented patterns more quickly.

Lines 165-168: These two sentences are fragments that could be integrated with one another to provide more clarity to the intended statement.

Line 187: Suggest changing “RTUs were recorded in all samples” to “RTUs were recorded from all samples” to more clearly indicate that it was a total number of RTUs from the entire experiment and not an equivalent number recovered from each severity treatment.

Table 3: Suggest removing the indent for Araneae and Pseudoscorpions and removing the “Arachnida” subheading. Acari are a taxon located within the Arachnida and the current format seems to suggest otherwise.

Lines 215-216: I think a note that “healthy” soils are being discussed is due to the fact that fungal hyphal networks are particularly susceptible to disturbance, and so the presence of such organisms demonstrates a more stable ecosystem.

Reviewer 2 Report

The manuscript investigates the impact of postfire on edaphic fauna, applying remote sensing methods to assign different levels of fire severity. However not only the few data, but also the scarce elaboration, make the article conclusions a little weak. The p value significance at p<0.1 make results a little inaccurate and, nevertheless, scarce.  The content of the manuscript is very interesting and of scientific interest, however could be improved.

INTRODUCTION

L.29-30: edaphic fauna size should be indicated

L.31: Myriapoda

L.35: one of the most

MATERIALS AND METHODS

L.105: the three sampling plots are considered replicates?

L.111: How much time passed between sampling and the beginning of the extraction?

L.112: Berlese-Tüllgren

L.116: Only alcohol?

L.118: titles are in italic

L.120-121: not only RTU was used, in table 3 different taxonomical levels were applied too.

L.131: while Jaccard and not Bray-Curtis? In this way you do not consider abundances but only presence.

No statistical tests were conducted to compare indexes, n° of taxa, abundance in the different levels of fire severity. Why? This would have improved your results and make conclusions more robust.

RESULTS AND DISCUSSION

L.193-195: why not testing it? There are functions that allow to test relationships between species/taxa and group of sites

L.214: Depends on the Collembola. They have a variety of feeding habits and morphologies and, as you wrote above, they are ubiquitous, so that not all of them indicate healthy soils.
